# Topological photonic band gaps in honeycomb atomic arrays

Pierre Wulles[1] and Sergey E. Skipetrov[1*]

**1** Univ. Grenoble Alpes, CNRS, LPMMC, 38000 Grenoble, France

* sergey.skipetrov@lpmmc.cnrs.fr

August 25, 2023

## Abstract

The spectrum of excitations a two-dimensional, planar honeycomb lattice of two-level atoms coupled by the in-plane electromagnetic field may exhibit band gaps that can be opened either by applying an external magnetic field or by breaking the symmetry between the two triangular sublattices of which the honeycomb one is a superposition. We establish the conditions of band gap opening, compute the width of the gap, and characterize its topological property by a topological index (Chern number). The topological nature of the band gap leads to inversion of the population imbalance between the two triangular sublattices for modes with frequencies near band edges. It also prohibits a transition to the trivial limit of infinitely spaced, noninteracting atoms without closing the spectral gap. Surrounding the lattice by a Fabry-Pérot cavity with small intermirror spacing $d < \pi/k_0$, where $k_0$ is the free-space wave number at the atomic resonance frequency, renders the system Hermitian by suppressing the leakage of energy out of the atomic plane without modifying its topological properties. In contrast, a larger $d$ allows for propagating optical modes that are built up due to reflections at the cavity mirrors and have frequencies inside the band gap of the free-standing lattice, thus closing the latter.

# 1 Introduction

Interaction of light with an isolated atom is the strongest when the frequency of light is close to one of the atomic resonance frequencies, corresponding to a transition between two quantum states of the atom [1]. In a dense ensemble of a large number $N$ of atoms, quantum states of individual atoms hybridize to give rise to collective atomic states [2,3]. Transitions of the atomic ensemble between these collective states result in collective resonances at frequencies that are different from those of isolated atoms [4,5]. The problem of calculating collective resonance frequencies of many-atom systems is particularly complicated when $N \gg 1$ and interatomic distances are of the order of the optical wavelength in free space $\lambda_0$. Remarkable progress has been achieved only under several simplifying assumptions. First, one assumes that all atoms are identical and that each atom has only two energy levels of which the first (generally, the lower one referred to as "ground state") is nondegenerate (total angular momentum $J_g = 0$) while the second (the upper one referred to as "excited state") is three-fold degenerate ($J_e = 1$). Second, the positions of atoms are assumed to be periodic in space giving rise to a structure known as "photonic crystal" [6]. Three-dimensional (3D) photonic crystals made of two-level atoms were predicted to give rise to photonic band gaps [7,8]. More recently, two-dimensional (2D) photonic crystals have been proposed as a useful playground for studying topological physics of light [9–12]. At the same time, relaxing any of the two above simplifying assumptions results in considerable complications. Whereas considering degenerate ground states leads to a need of dealing with large matrices of size growing exponentially with $N$, breaking down the periodicity of the atomic configuration by introducing randomness in atomic positions leads to such new and still poorly explored physical phenomena as Anderson localization of light [13,14] and photonic topological Anderson insulator [15].

In the present work we consider a 2D honeycomb lattice of two-level atoms coupled by the electromagnetic field. This system can be seen as a photonic analog of graphene or as an (artificial) "photonic graphene" for short [16]: carbon atoms are replaced by the two-level atoms that are now much further from each other (lattice spacing $a \sim$ tens of nm instead of $a \sim 1\text{Å}$ in graphene) whereas chemical bonds between nearest-neighbor carbon atoms are replaced by long-range electromagnetic coupling between all atoms via exchange of photons. Opening of a topological gap in the spectrum of such a lattice by an external, static magnetic field has been demonstrated in Ref. [9]. We extend the previous analysis in two different ways. First, we include the possibility of breaking the inversion symmetry between atoms of the two triangular sublattices that make up the honeycomb lattice. This provides possibilities of opening a topologically trivial band gap in the spectrum of the lattice and of studying the competition between the time-reversal (due to the magnetic field) and inversion symmetry breakings. Second, we study the effect of surrounding the atomic lattice by two plane-parallel reflecting plates forming a Fabry-Pérot cavity. The advantage of such a configuration resides in suppression of energy leakage out of the atomic plane, which makes the system Hermitian. In addition, it corresponds to an experimental setup used to study photonic topological phenomena with resonant dielectric scatterers in place of atoms [17]. Although dielectric scatterers differ from atomic ones in several respects (large size, existence of multiple electromagnetic resonances, insensitivity to the magnetic field), our results may still be useful for interpretation of certain microwave and optical experiments in dielectric systems under particular conditions.

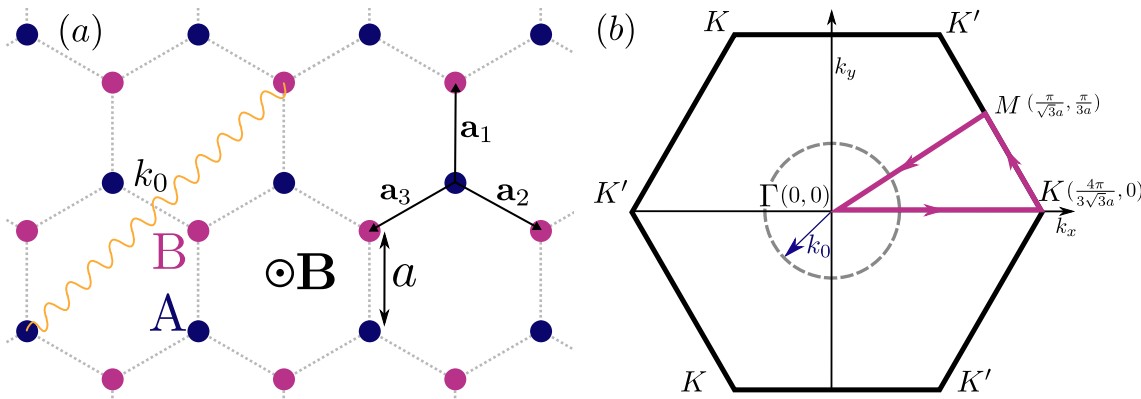

Figure 1: ($a$) Honeycomb lattice of two-level atoms with interatomic spacing $a$. The lattice can be seen as a superposition of two triangular sublattices $A$ (blue disks) and $B$ (red disks). The wavy line indicates coupling of each atom to all the others via the electromagnetic field. The lattice is placed in a static magnetic field **B** perpendicular to the atomic plane. $\mathbf{a}_1$, $\mathbf{a}_2$ and $\mathbf{a}_3$ are basis vectors of the lattice. ($b$) The first Brillouin zone of the honeycomb atomic lattice. The purple path is followed to construct the band diagram in Fig. 2. The dashed circle delimits the free-space light cone $|\mathbf{k}| < k_0$.

## 2 Honeycomb atomic lattice in the free space

### 2.1 The model

We consider a 2D honeycomb lattice of $N \gg 1$ atoms (interatomic spacing $a$) located at positions $\{\mathbf{r}_n\}$ ($n = 1, \ldots, N$) in the $xy$ plane $z = 0$, see Fig. 1. The lattice can be seen as a superposition of two triangular sublattices $A$ and $B$, with atoms $A$ and $B$ having single-atom ground and excited states $|g_n\rangle$ and $|e_{nm}\rangle$ ($m = 0, \pm 1$) with energies $E_g$ and $E_e^{(A,B)} = E_g + \hbar\omega_{A,B}$, respectively. Here we assume that the total angular momenta $\mathbf{J}_{g,e}$ of the ground and excited states have magnitudes $J_g = 0$ and $J_e = 1$ (in units of the Planck constant $\hbar$), respectively. Thus, the excited state of an isolated atom is triply degenerate, with the three substates corresponding to projections $m = 0, \pm 1$ of $\mathbf{J}_e$ on the quantization axis $z$. If $\omega_A \neq \omega_B$, then the inversion symmetry is broken. In addition, the time-reversal symmetry can be broken by an external magnetic field **B** perpendicular to the atomic plane. The ground state of the lattice is unique (nondegenerate) and corresponds to all atoms in their respective single-atom ground states. Propagation of a quasi-resonant excitation of the ground state can be described by an effective Hamiltonian obtained by extending the results of Refs. [9, 18] to include the possibility for atoms $A$ and $B$ to have different transition frequencies $\omega_A \neq \omega_B$ (see also Refs. [12, 15]):

$$\hat{\mathbb{H}} = \sum_{n=1}^{N} \sum_{m=-1}^{1} \left[ \hbar\omega_{A,B} + m g_e \mu_B |\mathbf{B}| - i\frac{\hbar\Gamma_0}{2} \right] |e_{nm}\rangle \langle e_{nm}|$$

$$+ \frac{3\pi\hbar\Gamma_0}{k_0} \sum_{n \neq n'} \sum_{m,m'=-1}^{1} \left[ \hat{d}_{eg} \hat{\mathcal{G}}(\mathbf{r}_n, \mathbf{r}_{n'}) \hat{d}_{eg}^\dagger \right]_{mm'} |e_{nm}\rangle \langle e_{n'm'}| \tag{1}$$

where $k_0 = \omega_0/c$, $\omega_0 = (\omega_A + \omega_B)/2$, $c$ is the speed of light in the free space, $\mu_B$ is the Bohr magneton, $g_e$ is the Landé factor of the excited states (the same for atoms $A$ and $B$), $g_e \mu_B |\mathbf{B}|$ is the Zeeman shift of energies of the single-atom excited states, $\Gamma_0$ is the radiative line width of an individual atom in the free space and $\hat{\mathcal{G}}$ is the dyadic Green's function describing the

coupling of atoms by electromagnetic waves:

$$
\begin{aligned}
\hat{\mathcal{G}}(\mathbf{r}, \mathbf{r'}) &= \hat{\mathcal{G}}(\mathbf{r} - \mathbf{r'}) \\
&= \frac{\delta(\mathbf{r} - \mathbf{r'})}{3k_0^2}\mathbb{1}_3 - \frac{e^{ik_0|\mathbf{r}-\mathbf{r'}|}}{4\pi|\mathbf{r}-\mathbf{r'}|}\left[P(ik_0|\mathbf{r}-\mathbf{r'}|)\mathbb{1}_3 + Q(ik_0|\mathbf{r}-\mathbf{r'}|)\frac{(\mathbf{r}-\mathbf{r'})\otimes(\mathbf{r}-\mathbf{r'})}{(\mathbf{r}-\mathbf{r'})^2}\right] \quad (2)
\end{aligned}
$$

with $\mathbb{1}_3$ the $3 \times 3$ unit matrix, $P(u) = 1 - 1/u + 1/u^2$ and $Q(u) = -1 + 3/u - 3/u^2$. Equation (1) also makes use of the matrix

$$
\hat{d}_{eg} = \begin{bmatrix} \frac{1}{\sqrt{2}} & \frac{i}{\sqrt{2}} & 0 \\ 0 & 0 & 1 \\ -\frac{1}{\sqrt{2}} & \frac{i}{\sqrt{2}} & 0 \end{bmatrix} \quad (3)
$$

that transforms the Green's function (2) into the basis of circular polarizations in the $xy$ plane while leaving the $z$ component intact.

For atoms in the plane $z = 0$, the Green's function (2) does not couple in-plane excitations (i.e., those having only $x$ and $y$ components) to out-of-plane ones (i.e., those having only the $z$ component). In addition, the magnetic field has no effect on the $m = 0$ magnetic substate. Thus, we consider only in-plane excitations (often referred to as transverse-electric or TE for short) from here on and represent the state of the lattice by a vector $|\Psi\rangle$ composed of $N/2$ 4-component spinors $|\Psi_n\rangle = \{\Psi_n^{A+}, \Psi_n^{A-}, \Psi_n^{B+}, \Psi_n^{B-}\}^T$, where $\Psi_n^{A\pm}$ is the value of the wave function on the atom $A$ of the elementary cell $n$ composed of a pair of atoms $A$, $B$ ($n = 1, \ldots, N/2$) for $m = \pm 1$, respectively (and similarly for $\Psi_n^{B\pm}$). The problem then reduces to the analysis of a $2N \times 2N$ non-Hermitian effective Hamiltonian composed of $4 \times 4$ blocks [9, 12]

$$
\begin{aligned}
\hat{H}_{nn'} &= \delta_{nn'}\left\{\begin{bmatrix} i\text{Im}G_{mm}(0)\mathbb{1}_2 & \hat{G}(-\mathbf{a}_1) \\ \hat{G}(\mathbf{a}_1) & i\text{Im}G_{mm}(0)\mathbb{1}_2 \end{bmatrix} + 2\Delta_{AB}\begin{bmatrix} \mathbb{1}_2 & 0 \\ 0 & -\mathbb{1}_2 \end{bmatrix} + 2\Delta_{\mathbf{B}}\begin{bmatrix} \hat{\sigma}_z & 0 \\ 0 & \hat{\sigma}_z \end{bmatrix}\right\} \\
&\quad + (1 - \delta_{nn'})\begin{bmatrix} \hat{G}(\mathbf{r}_n - \mathbf{r}_{n'}) & \hat{G}(\mathbf{r}_n - \mathbf{r}_{n'} - \mathbf{a}_1) \\ \hat{G}(\mathbf{r}_n - \mathbf{r}_{n'} + \mathbf{a}_1) & \hat{G}(\mathbf{r}_n - \mathbf{r}_{n'}) \end{bmatrix} \quad (4)
\end{aligned}
$$

where $\mathbf{r}_n$ and $\mathbf{r}_n + \mathbf{a}_1$ are positions of atoms $A$ and $B$ of the unit cell $n$, $\mathbb{1}_2$ is the $2 \times 2$ unit matrix, $\hat{\sigma}_z$ is the third Pauli matrix,

$$
\hat{G}(\mathbf{r}) = -\frac{6\pi}{k_0}\hat{d}_{2D}\hat{\mathcal{G}}_{\{2\}}(\mathbf{r})\hat{d}_{2D}^\dagger \quad (5)
$$

$\hat{\mathcal{G}}_{\{2\}}(\mathbf{r})$ is the leading principal submatrix of order 2 [i.e., the $2 \times 2$ matrix in the upper-left corner of the $3 \times 3$ matrix $\hat{\mathcal{G}}(\mathbf{r})$], $\Delta_{\mathbf{B}} = g_e\mu_B|\mathbf{B}|/\hbar\Gamma_0$, $\Delta_{AB} = (\omega_B - \omega_A)/2\Gamma_0$ and

$$
\hat{d}_{2D} = \frac{1}{\sqrt{2}}\begin{bmatrix} 1 & i \\ -1 & i \end{bmatrix} \quad (6)
$$

Complex eigenavalues $\Lambda$ of $\hat{H}$ yield frequencies $\omega$ and decay rates $\Gamma$ of collective excitations in the atomic lattice: $\Lambda = -2(\omega - \omega_0)/\Gamma_0 + i\Gamma/\Gamma_0$.

Diagonal elements of the first term in Eq. (4) contain the imaginary part of the Green's function at the origin. It is proportional to the local density of states and describes the decay rate of an isolated atom. $\hat{G}(\mathbf{r})$ is normalized in such a way that $\text{Im}G_{mm}(0) = 1$ with $m = \pm 1$ [$G_{++}(0) = G_{--}(0)$ in the free space]. This corresponds to the decay rate $\Gamma = \Gamma_0$. $G_{mm}(0)$ also has a divergent real part that corresponds to a frequency shift (Lamb shift) due to the interaction of the atom with the electromagnetic vacuum. In our formalism, it is supposed to be already included in the definition of the atomic transition frequency $\omega_0$ and thus we do not include it explicitly in the Hamiltonian (4).

## 2.2  Band diagram of a honeycomb lattice in the free space

For the infinite lattice without boundaries $N \to \infty$ and we can use Bloch theorem to look for the components $|\Psi_n\rangle$ of $|\Psi\rangle$ in the form $|\Psi_n\rangle = |\psi(\mathbf{k})\rangle e^{i\mathbf{k}\cdot\mathbf{r}_n}$. Schrödinger equation $\hat{H}|\Psi\rangle = \Lambda|\Psi\rangle$ then reduces to

$$\hat{\mathcal{H}}(\mathbf{k})|\psi(\mathbf{k})\rangle = \Lambda(\mathbf{k})|\psi(\mathbf{k})\rangle \tag{7}$$

where

$$\hat{\mathcal{H}}(\mathbf{k}) = \begin{bmatrix} \mathcal{H}_{11}^{++} & \mathcal{H}_{11}^{+-} & \mathcal{H}_{12}^{++} & \mathcal{H}_{12}^{+-} \\ \mathcal{H}_{11}^{-+} & \mathcal{H}_{11}^{--} & \mathcal{H}_{12}^{-+} & \mathcal{H}_{12}^{--} \\ \mathcal{H}_{21}^{++} & \mathcal{H}_{21}^{+-} & \mathcal{H}_{22}^{++} & \mathcal{H}_{22}^{+-} \\ \mathcal{H}_{21}^{-+} & \mathcal{H}_{21}^{--} & \mathcal{H}_{22}^{-+} & \mathcal{H}_{22}^{--} \end{bmatrix} \tag{8}$$

and

$$\hat{\mathcal{H}}_{11}(\mathbf{k}) = \sum_{\mathbf{r}_n \neq 0} \hat{G}(\mathbf{r}_n) e^{i\mathbf{k}\cdot\mathbf{r}_n} + \begin{bmatrix} i + 2\Delta_{AB} + 2\Delta_{\mathbf{B}} & 0 \\ 0 & i + 2\Delta_{AB} - 2\Delta_{\mathbf{B}} \end{bmatrix} \tag{9}$$

$$\hat{\mathcal{H}}_{12}(\mathbf{k}) = \sum_{\mathbf{r}_n} \hat{G}(\mathbf{r}_n + \mathbf{a}_1) e^{i\mathbf{k}\cdot\mathbf{r}_n} \tag{10}$$

$$\hat{\mathcal{H}}_{21}(\mathbf{k}) = \sum_{\mathbf{r}_n} \hat{G}(\mathbf{r}_n - \mathbf{a}_1) e^{i\mathbf{k}\cdot\mathbf{r}_n} \tag{11}$$

$$\hat{\mathcal{H}}_{22}(\mathbf{k}) = \sum_{\mathbf{r}_n \neq 0} \hat{G}(\mathbf{r}_n) e^{i\mathbf{k}\cdot\mathbf{r}_n} + \begin{bmatrix} i - 2\Delta_{AB} + 2\Delta_{\mathbf{B}} & 0 \\ 0 & i - 2\Delta_{AB} - 2\Delta_{\mathbf{B}} \end{bmatrix} \tag{12}$$

are $2 \times 2$ matrices.

Summations in Eqs. (9–12) are most efficiently perfformed in momentum space using Poisson's summation formula [7, 9]:

$$\sum_{\mathbf{r}_n \neq 0} \hat{G}(\mathbf{r}_n) e^{i\mathbf{k}\cdot\mathbf{r}_n} = \frac{1}{\mathcal{A}} \sum_{\mathbf{g}_m} \hat{g}(\mathbf{g}_m - \mathbf{k}) - \hat{G}(\mathbf{r} = 0) \tag{13}$$

where $\mathcal{A}$ is the area of the unit cell of the lattice, $\mathbf{g}_m$ are 2D reciprocal lattice vectors obeying $\mathbf{g}_m \cdot \mathbf{r}_n = 2\pi m$ with an integer $m$,

$$\hat{g}(\mathbf{q}_\perp) = \int\limits_{-\infty}^{\infty} \frac{dq_z}{2\pi} \hat{G}(\mathbf{q}) \tag{14}$$

$\mathbf{q}_\perp = \{q_x, q_y\}$ is the in-plane component of the 3D vector $\mathbf{q}$: $\mathbf{q} = \{\mathbf{q}_\perp, q_z\}$ and $\hat{G}(\mathbf{q})$ is Fourier transform of $\hat{G}(\mathbf{r})$:

$$\hat{G}(\mathbf{q}) = -\frac{6\pi}{k_0} \hat{d}_{2D} \hat{\mathcal{G}}_{\{2\}}(\mathbf{q}) \hat{d}_{2D}^\dagger \tag{15}$$

$$\hat{\mathcal{G}}(\mathbf{q}) = \int d^3\mathbf{r}\, \hat{\mathcal{G}}(\mathbf{r}) e^{i\mathbf{q}\cdot\mathbf{r}} = \frac{\mathbf{q} \otimes \mathbf{q}}{q^2 k_0^2} + \frac{\mathbb{1} - (\mathbf{q} \otimes \mathbf{q})/q^2}{k_0^2 - q^2 + i0^+} = \frac{\mathbb{1} - (\mathbf{q} \otimes \mathbf{q})/k_0^2}{k_0^2 - q^2 + i0^+} \tag{16}$$

Similarly,

$$\sum_{\mathbf{r}_n} \hat{G}(\mathbf{r}_n \pm \mathbf{a}_1) e^{i\mathbf{k}\cdot\mathbf{r}_n} = \frac{1}{\mathcal{A}} \sum_{\mathbf{g}_m} \hat{g}(\mathbf{g}_m - \mathbf{k}) e^{\pm i\mathbf{a}_1 \cdot (\mathbf{g}_m - \mathbf{k})} \tag{17}$$

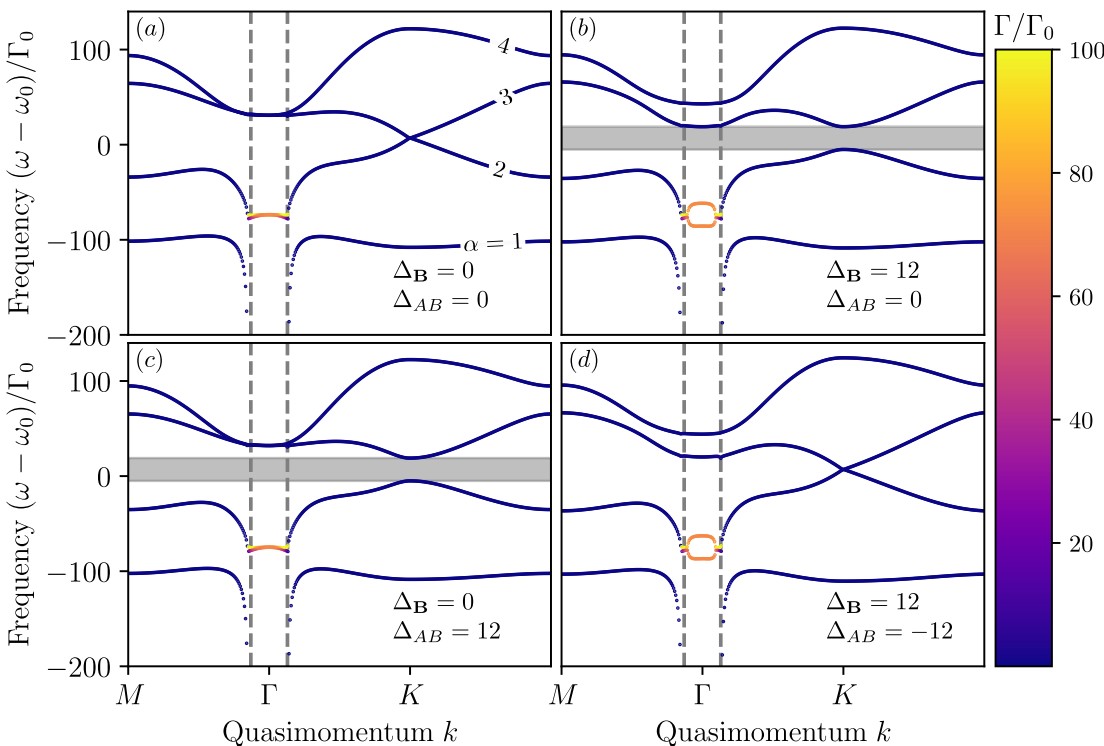

Figure 2: Band diagrams of a honeycomb lattice of atoms coupled via the electromagnetic field for $k_0 a = 2\pi \times 0.05$ [see Fig. 1(a)] and four different pairs of $\Delta_{\mathbf{B}}$, $\Delta_{AB}$. Horizontal axis corresponds to the path shown in purple in the 2D Brillouin zone in Fig. 1(b). Color code corresponds to the decay rate $\Gamma$ of quasimodes capped at $\Gamma/\Gamma_0 = 100$ (i.e., all $\Gamma/\Gamma_0 \geq 100$ are shown in yellow). We label bands from bottom to top by an index $\alpha = 1$–4. Gray shaded areas are band gaps between the second and third spectral bands, gray dashed lines delimit the free-space light cone $|\mathbf{k}| \leq k_0$ where $\Gamma > 0$.

The integral in Eq. (14) and $\hat{G}(\mathbf{r} = 0)$ both diverge but these divergences cancel out in Eq. (13). To compute each of these two divergent terms separately, we introduce Gaussian cut-offs $1/h$ in the momentum integrals for $\hat{g}(\mathbf{q})$ and $\hat{G}(\mathbf{r} = 0)$ and replace the latter by their regularized versions

$$
\hat{g}^{(\mathrm{reg})}(\mathbf{q}_\perp) = \int_{-\infty}^{\infty} \frac{dq_z}{2\pi} \hat{G}(\mathbf{q}) e^{-h^2 \mathbf{q}^2/2}
$$

$$
= \frac{3\pi}{k_0} \frac{e^{-k_0^2 h^2/2}}{\sqrt{k_0^2 - q_\perp^2}} \left[ \mathbb{1}_2 - \frac{\hat{d}_{2D}(\mathbf{q}_\perp \otimes \mathbf{q}_\perp)\hat{d}_{2D}^\dagger}{k_0^2} \right] \left[ i - \mathrm{erfi}\left( \frac{h}{2} \sqrt{k_0^2 - q_\perp^2} \right) \right] \quad (18)
$$

$$
\hat{G}^{(\mathrm{reg})}(\mathbf{r} = 0) = \int \frac{d^3\mathbf{q}}{(2\pi)^3} \hat{G}(\mathbf{q}) e^{-h^2 \mathbf{q}^2/2}
$$

$$
= \left\{ \left[ i - \mathrm{erfi}\left( k_0 h/\sqrt{2} \right) \right] e^{-k_0^2 h^2/2} - \frac{1/2 - (k_0 h)^2}{\sqrt{\pi/2}(k_0 h)^3} \right\} \mathbb{1}_2 \quad (19)
$$

In practice, we use Eqs. (18) and (19) with a small $h < 0.1a$ and sum over a sufficiently large number of reciprocal lattice vectors $\mathbf{g}_m$ in Eqs. (13) and (17) to ensure convergence. Using a smaller $h$ ensures better accuracy of results but requires taking into account a larger number

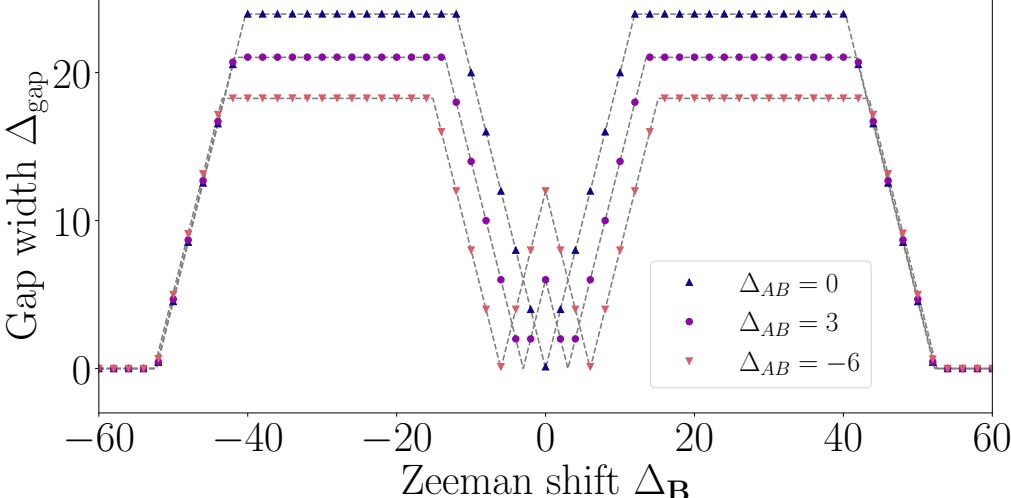

Figure 3: Width of the gap between the second and third bands in the band diagram of Fig. 2 as a function of $\Delta_{\mathbf{B}}$ for three different values of $\Delta_{AB}$ and fixed $k_0 a = 2\pi \times 0.05$. Symbols show $\Delta_{\mathrm{gap}}$ determined via numerical diagonalization of the Hamiltonian $\hat{\mathcal{H}}(\mathbf{k})$, gray dashed lines represent the analytical result (20).

of $\mathbf{g}_m$ for convergence of sums in Eqs. (13) and (17).

We show representative examples of band diagrams obtained by diagonalizing $\hat{\mathcal{H}}(\mathbf{k})$ in Fig. 2. In the absence of symmetry breaking $\Delta_{\mathbf{B}} = \Delta_{AB} = 0$, two of the four bands—the second and third ones—cross at a degeneracy point $K$ [see Fig. 2(a)] where dispersion is roughly linear and $(\omega - \omega_0)/\Gamma_0 \simeq 7$. The same crossing takes place at the second degeneracy point $K'$ [see Fig. 1(b)] and three equivalent pairs of points $K$, $K'$ exist in the Brillouin zone (BZ), exactly as in the electronic band diagram of graphene [19]. Breaking the time-reversal ($\Delta_{\mathbf{B}} \neq 0$) or inversion ($\Delta_{AB} \neq 0$) symmetry opens a gap between the second and third bands of the spectrum as we illustrate in Figs. 2(b) and (c). However, the gap closes if $|\Delta_{\mathbf{B}}| = |\Delta_{AB}|$ [see Fig. 2(d)]. Analysis shows that the gap closing takes place either at $K$ point when $\Delta_{\mathbf{B}} = -\Delta_{AB}$ as in Fig. 2(d) or at $K'$ point when $\Delta_{\mathbf{B}} = \Delta_{AB}$ (not shown).

For $\mathbf{k}$ inside the light cone $|\mathbf{k}| < k_0$ [see Fig. 1(b)], eigenvalues of $\hat{\mathcal{H}}$ acquire an imaginary part $\Gamma/\Gamma_0$ shown in Fig. 2 by a color code. This is due to the possibility of electromagnetic wave emission out of the atomic plane $z = 0$. As follows from Fig. 2, the decay rate $\Gamma$ of collective excitations corresponding to $|\mathbf{k}| < k_0$ can exceed the decay rate $\Gamma_0$ of an isolated atom by several orders of magnitude. This phenomenon is known as superradiance: many atoms synchronize to emit energy very rapidly, during a time interval of the order of $1/\Gamma \ll 1/\Gamma_0$ [20]. The atomic array becomes an efficient source of light in this regime. Note that momentum conservation forbids the emission of electromagnetic waves out of the atomic plane $z = 0$ for $\mathbf{k}$ outside the light cone (i.e., for $|\mathbf{k}| > k_0$), and the eigenvalues $\Lambda(\mathbf{k})$ of $\hat{\mathcal{H}}(\mathbf{k})$ are real in this part of the band diagram.

## 2.3 Width of the band gap

The width of the spectral gap $\Delta_{\mathrm{gap}}$ between the second and third bands in Fig. 2 is controlled by three parameters: $\Delta_{\mathbf{B}}$, $\Delta_{AB}$, and $k_0 a$. It can be found by analyzing the Hamiltonian $\hat{\mathcal{H}}(\mathbf{k})$ at $K$, $K'$ and $\Gamma$ points of BZ. We provide the details of derivations in Appendix A and summarize here the main results. Let us first consider the simplest case of $\Delta_{AB} = 0$. Four regimes can be distinguished depending on the value of $|\Delta_{\mathbf{B}}|$. Defining three positive threshold values of $|\Delta_{\mathbf{B}}|$:

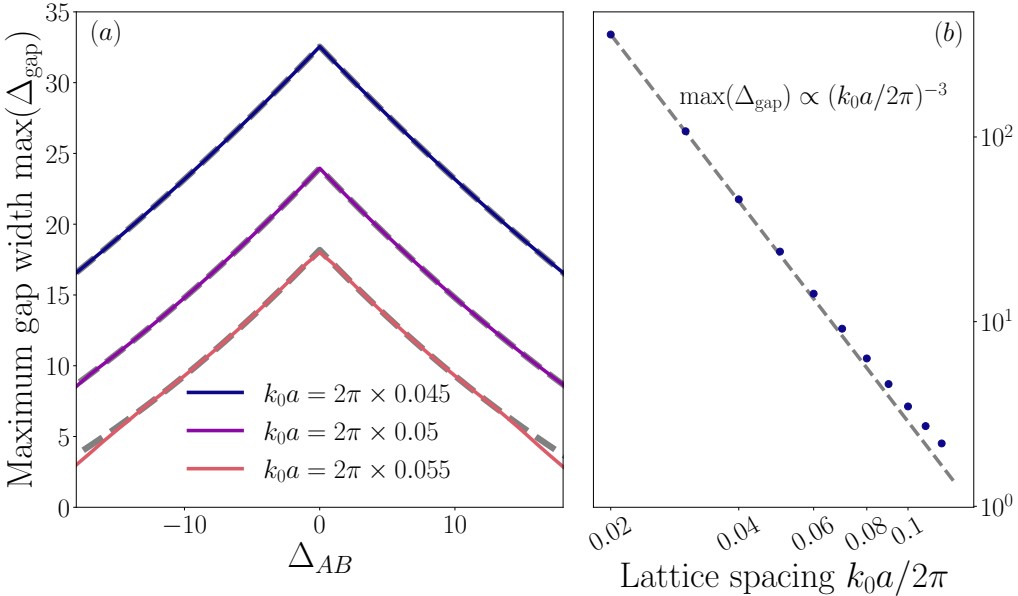

Figure 4: (*a*) Maximum width of the gap $\max(\Delta_{\text{gap}})$ between the second and third bands in the band diagram of Fig. 2 as a function of $\Delta_{AB}$ for three different values of $k_0 a$. Solid lines show numerical results obtained for $\Delta_{\mathbf{B}} = 30$. Dashed lines correspond to the formula in the second line of Eq. (20). (*b*) Log-log plot of $\max(\Delta_{\text{gap}})$ as a function of $k_0 a$ at $\Delta_{AB} = 0$. Symbols show numerical results obtained for $\Delta_{\mathbf{B}} = 12$. Gray dashed line is a power-law fit $\max(\Delta_{\text{gap}}) = A(k_0 a/2\pi)^{-3}$ for $k_0 a/2\pi < 0.05$ with the best-fit value $A = 3.24 \times 10^{-3}$.

$\Delta_{\mathbf{B}}^{(1)} < \Delta_{\mathbf{B}}^{(2)} < \Delta_{\mathbf{B}}^{(3)}$, we identify the following scenario of gap width evolution as $|\Delta_{\mathbf{B}}|$ increases (see triangles in Fig. 3 for an illustration at $k_0 a = 2\pi \times 0.05$). First, for small $|\Delta_{\mathbf{B}}|$, a direct gap opens at $K$ and $K'$ points and its width increases linearly with $|\Delta_{\mathbf{B}}|$ for $|\Delta_{\mathbf{B}}| < \Delta_{\mathbf{B}}^{(1)}$. Next, the gap becomes indirect and its width $\Delta_{\text{gap}}$ is determined by the frequency difference between the frequency of the second band at $K$ (or $K'$) point and the frequency of the third band at $\Gamma$ point. $\Delta_{\text{gap}}$ remains constant in this regime until $|\Delta_{\mathbf{B}}|$ reaches $\Delta_{\mathbf{B}}^{(2)}$. Starting from this value of $|\Delta_{\mathbf{B}}|$ the gap becomes direct again and is equal to the frequency difference between the second and third bands at $\Gamma$ point. Its width decreases linearly with $|\Delta_{\mathbf{B}}|$ until gap closing for $|\Delta_{\mathbf{B}}| = \Delta_{\mathbf{B}}^{(3)}$. The gap remains closed ($\Delta_{\text{gap}} = 0$) for $|\Delta_{\mathbf{B}}| > \Delta_{\mathbf{B}}^{(3)}$.

The above scenario can be generalized to the case of nonzero but still moderate $\Delta_{AB}$ (see Appendix A). The final result for the gap width is

$$\Delta_{\text{gap}} = 2 \times \begin{cases} ||\Delta_{\mathbf{B}}| - |\Delta_{AB}||, & |\Delta_{\mathbf{B}}| < \Delta_{\mathbf{B}}^{(1)} \\ |\Delta_{\mathbf{B}}^{(1)} - |\Delta_{AB}||, & \Delta_{\mathbf{B}}^{(1)} < |\Delta_{\mathbf{B}}| < \Delta_{\mathbf{B}}^{(2)} \\ |\Delta_{\mathbf{B}}^{(1)} - |\Delta_{AB}|| + \Delta_{\mathbf{B}}^{(2)} - |\Delta_{\mathbf{B}}|, & \Delta_{\mathbf{B}}^{(2)} < |\Delta_{\mathbf{B}}| < \Delta_{\mathbf{B}}^{(3)} \\ 0, & |\Delta_{\mathbf{B}}| > \Delta_{\mathbf{B}}^{(3)} = |\Delta_{\mathbf{B}}^{(1)} - |\Delta_{AB}|| + \Delta_{\mathbf{B}}^{(2)} \end{cases} \tag{20}$$

As follows from the derivations in Appendix A, the parameters $\Delta_{\mathbf{B}}^{(n)}$ are functions of $k_0 a$ and $|\Delta_{AB}|$. Figure 3 illustrates the very good agreement between Eq. (20) shown by dashed lines and numerical results (symbols).

From the point of view of experimental observation of phenomena discussed in this work, of particular importance is the maximum width of the gap that can exist in the considered atomic system. As follows from Fig. 3, the maximum of $\Delta_{\text{gap}}$ is reached when the latter

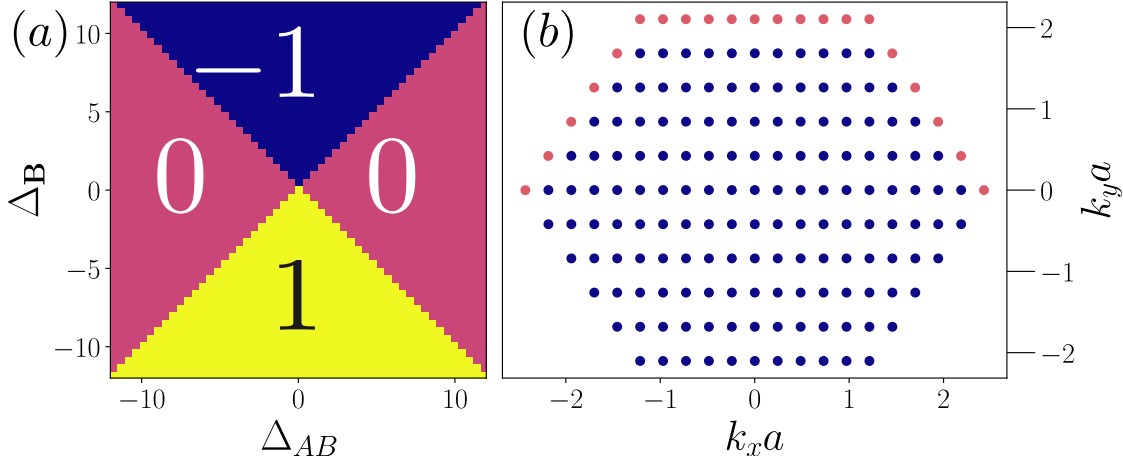

Figure 5: (*a*) Chern number of the spectral gap $C$ as a function of $\Delta_{\mathbf{B}}$ and $\Delta_{AB}$ for $k_0 a = 2\pi \times 0.05$: $C = 0$ means that the gap is trivial, $C = \pm 1$ means that the gap is topological. The set of discrete values $\Delta_{\mathbf{B}}$, $\Delta_{AB}$ used to create this graph is chosen to avoid the exact equality $|\Delta_{\mathbf{B}}| = |\Delta_{AB}|$ for which there is no spectral gap between the second and third bands in Fig. 2(a) and the two bands cannot be separated unambiguously. (*b*) Discrete lattice of points $\mathbf{k}_n$ used to compute the Chern number $C_\alpha$ of spectral bands. Summation in Eq. (26) is performed over $\mathbf{k}_n$ corresponding to blue dots whereas red dots belong to adjacent Brillouin zones and are not included in the sum.

plateaus for $\Delta_{\mathbf{B}}^{(1)} < |\Delta_{\mathbf{B}}| < \Delta_{\mathbf{B}}^{(2)}$. We show the maximum gap width $\max(\Delta_{\text{gap}})$ as a function of $\Delta_{AB}$ for several values of $k_0 a$ and as a function of $k_0 a$ for $\Delta_{AB} = 0$ in Figs. 4(a) and (b), respectively. Numerical results are compared to the second line of Eq. (20) in Fig. 4(a) and are fitted by a power law $\max(\Delta_{\text{gap}}) \propto (k_0 a)^{-3}$ in Fig. 4(b). The fact that the latter fit is of good quality has been already noticed in Ref. [9]. It testifies of the importance of near-field, dipole-dipole coupling between atoms for gap opening since the same scaling $(k_0 r)^{-3}$ is a characteristic feature of the Green's function (2) for $k_0 r \ll 1$.

## 2.4 Topological properties of the band structure

The band structures shown in Figs. 2(b) and (c) look very similar and the widths of the band gaps are exactly the same. However, there is a profound difference between these band structures, which can be made explicit by studying their topological properties. To this end, we compute the Chern number $C$ of a band $\alpha$ ($\alpha = 1$–4, see Fig. 2) as

$$C_\alpha = \frac{1}{2\pi} \int_{\text{BZ}} \Omega_\alpha(\mathbf{k}) d^2\mathbf{k} \tag{21}$$

with the Berry curvature

$$\Omega_\alpha(\mathbf{k}) = i\left[\left\langle \frac{\partial \psi_\alpha(\mathbf{k})}{\partial k_x} \bigg| \frac{\partial \psi_\alpha(\mathbf{k})}{\partial k_y} \right\rangle - \left\langle \frac{\partial \psi_\alpha(\mathbf{k})}{\partial k_y} \bigg| \frac{\partial \psi_\alpha(\mathbf{k})}{\partial k_x} \right\rangle\right] \tag{22}$$

Direct numerical evaluation of the integral in Eq. (21) by discretizing BZ faces several difficulties discussed, in particular, in Ref. [21]. To circumvent them, one introduces a rectangular lattice of points $\mathbf{k}_n = \{k_{nx}, k_{ny}\}$ that cover BZ [see Fig. 5(b)] and defines link variables [21]

$$U_\alpha^{(\mu)}(\mathbf{k}_n) = \frac{\langle \psi_\alpha(\mathbf{k}_n)|\psi_\alpha(\mathbf{k}_n + \mathbf{u}_\mu)\rangle}{|\langle \psi_\alpha(\mathbf{k}_n)|\psi_\alpha(\mathbf{k}_n + \mathbf{u}_\mu)\rangle|}, \quad \mu = x, y, \tag{23}$$

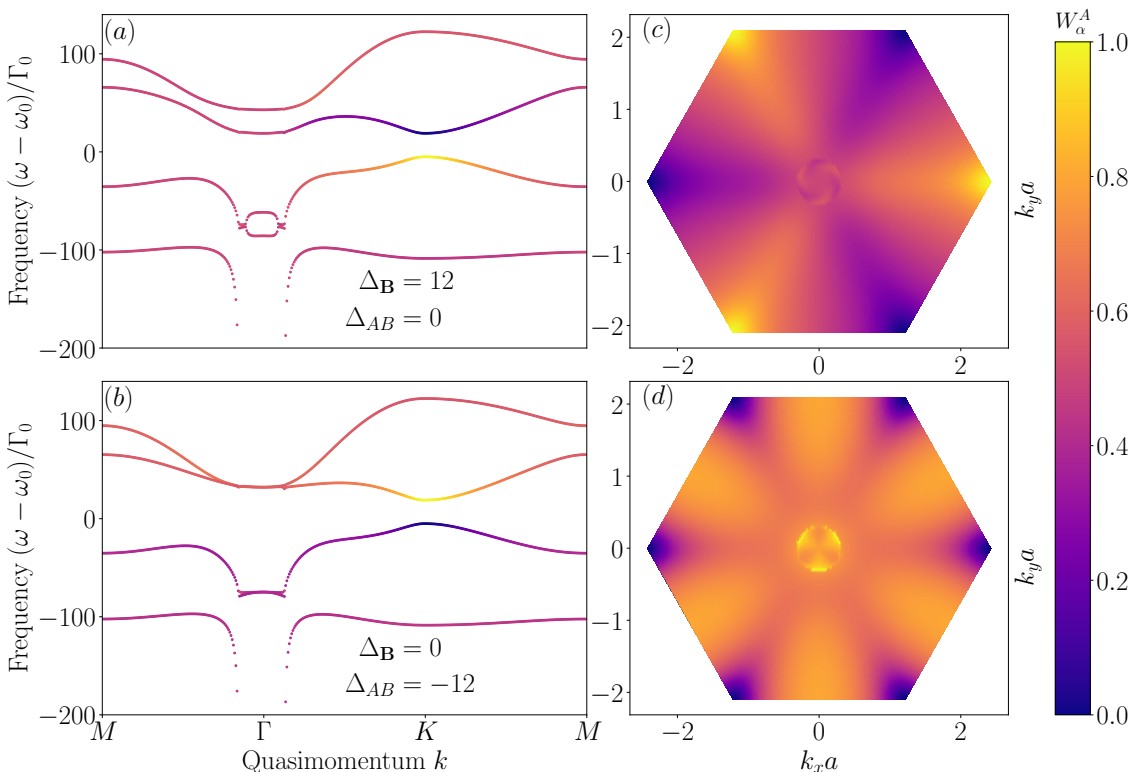

Figure 6: Topological (a) and trivial (b) band structures color-coded as a function of the weight of the quasimodes on the atomic sublattice $A$, $W_\alpha^A(\mathbf{k})$. Horizontal axes correspond to the path shown in purple in the 2D Brillouin zone in Fig. 1(b). ($c$) and ($d$) show the second band as a function of $\mathbf{k}$ in the 2D Brillouin zone for band diagrams in panels (a) and (b), respectively.

where $\mathbf{u}_\mu$ is a vector connecting the neighboring points of the lattice along $\mu$ axis. A lattice field strength is defined by

$$F_\alpha(\mathbf{k}_n) = \ln\left[U_\alpha^{(x)}(\mathbf{k}_n)U_\alpha^{(y)}(\mathbf{k}_n + \mathbf{u}_x)U_\alpha^{(x)}(\mathbf{k}_n + \mathbf{u}_y)^{-1}U_\alpha^{(y)}(\mathbf{k}_n)^{-1}\right] \tag{24}$$

$$-\pi < \frac{1}{i}F_\alpha(\mathbf{k}_n) \le \pi \tag{25}$$

Finally, the Chern number is [21]

$$C_\alpha = \frac{1}{2\pi i}\sum_n F_\alpha(\mathbf{k}_n). \tag{26}$$

It can be demonstrated that $C_\alpha$ defined by Eq. (26) takes integer values and coincides with the result of the direct evaluation of the Chern number using Eq. (21) provided that the lattice in $\mathbf{k}$ samples BZ finely enough [21]. The advantage of Eq. (26) with respect to the direct numerical integration of Eq. (21) is that it yields proper results at much larger discretization steps $u_x$, $u_y$ than those needed for accurate numerical integration in Eq. (21) and does not require specifying a particular gauge. Therefore, Eq. (26) allows for an accurate calculation of Chern numbers with a reasonable numerical effort.

Topological properties of a spectral gap can be characterized by the Chern number of the gap $C$ equal to the sum of $C_\alpha$ for the two bands below the gap: $C = C_1 + C_2$ if the bands are numbered starting from the bottom as shown in Fig. 2. We fix $k_0 a = 2\pi \times 0.05$ as in Fig. 2 and compute $C$ as a function of $\Delta_\mathbf{B}$ and $\Delta_{AB}$. The results are shown in Fig. 5(a). We

see that the gap is topological when $|\Delta_{\mathbf{B}}| > |\Delta_{AB}|$ and trivial otherwise. This condition is quite remarkable because the two parameters $\Delta_{\mathbf{B}}$ and $\Delta_{AB}$ measure the strengths of, respectively, time-reversal and inversion symmetries breakdowns. Thus, the gap in the spectrum of the atomic lattice turns out to be topologically nontrivial if and only if the breakdown of the time-reversal symmetry (quantified by $|\Delta_{\mathbf{B}}|$) is stronger than the breakdown of the inversion symmetry (quantified by $|\Delta_{AB}|$).

Some insight into topological properties of the band structure can be obtained even without computing topological indices by exploring the relative weight of quasimodes on sites of sublattices $A$ and $B$ (mode "polarization"). In our lattice, the weights of the wave function $\psi_\alpha(\mathbf{k})$ on sublattices $A$ and $B$ are

$$W_\alpha^A(\mathbf{k}) = \left|\psi_\alpha^{A+}(\mathbf{k})\right|^2 + \left|\psi_\alpha^{A-}(\mathbf{k})\right|^2 \tag{27}$$

$$W_\alpha^B(\mathbf{k}) = \left|\psi_\alpha^{B+}(\mathbf{k})\right|^2 + \left|\psi_\alpha^{B-}(\mathbf{k})\right|^2 \tag{28}$$

with a normalization condition $W_\alpha^A(\mathbf{k}) + W_\alpha^B(\mathbf{k}) = 1$. We show typical band diagrams color-coded as functions of $W_\alpha^A(\mathbf{k})$ in Figs. 6(a) and (b).

The bottom ($\alpha = 1$) and top ($\alpha = 4$) bands exhibit $W_\alpha^A(\mathbf{k}) \sim 0.5$ without any significant dependence on $\mathbf{k}$. In contrast, the second and third bands show interesting and opposite polarizations in the vicinity of $K$ point. More precisely, when the band gap is trivial as in Fig. 6(b), the excitations of the second (lower with respect to the spectral gap) band are localized on the atoms of $B$ sublattice whereas the excitations of the third (upper) band are localized on the atoms of $A$ sublattice. This property is common for both $K$ and $K'$ points of BZ as can be seem from Fig. 6(d). In contrast, when the band gap is topological as in Fig. 6(a), a band inversion phenomenon takes place: the second (lower with respect to the spectral gap) band is now due to excitations localized on the sublattice $A$ whereas the third (upper) band—to excitations localized on the sublattice $B$. For $\Delta_{\mathbf{B}} > 0$, band inversion takes place only in $K$ points of BZ whereas $K'$ points exhibit the same band polarization as in the topologically trivial case, see Fig. 6(c). The situation is the opposite for $\Delta_{\mathbf{B}} < 0$ (not shown in Fig. 6).

Whereas modifications of band polarization due to the opening of a topological band gap and band inversion are common for topologically nontrivial systems (see, e.g., Refs. [22–25] for examples in topological photonics), Figs. 6(c) and (d) also exhibit another interesting difference between $W_\alpha^A(\mathbf{k})$ for the topological (c) and trivial (d) band structures inside the light cone $|\mathbf{k}| < k_0$. Namely, Fig. 6(c) features a spiral structure for $|\mathbf{k}| < k_0$, which is very different from the three-lobe structure in Fig. 6(d). Because $|\mathbf{k}| < k_0$ corresponds to leaky quasimodes, this part of the band diagram should have visible consequences on the properties of light emitted by the atomic lattice. In particular, the spiral structure of Fig. 6(c) may lead to emission of light with a non-zero angular momentum, similarly to the emission of circularly polarized light by topologically nontrivial modes observed in Ref. [26] for a different photonic system.

# 3 Honeycomb atomic lattice in a Fabry-Pérot cavity

In Sec. 2, we have considered a 2D atomic lattice suspended in the 3D free space. The effective Hamiltonian describing such a lattice is non-Hermitian and thus the formal application of standard methods developed to characterize topological properties of excitations in Hermitian systems (see Sec. 2.4) may be questioned. In addition, we show that the decay rate $\Gamma$ of quasimodes inside the light cone $|\mathbf{k}| < k_0$ can be very high (see Fig. 2). If we interpret $\Gamma$ as an uncertainty of $\omega$, gaps in the spectrum of the lattice in Fig. 2 may become undetectable in practice. All these problems can be eliminated by placing the atomic lattice inside a Fabry-Pérot

cavity that cuts the leakage of energy out of the atomic system and renders the Hamiltonian Hermitian. Placing resonant scatterers (though not atoms) between plane-parallel reflecting plates is also a common strategy in microwave experiments in the field of topological photonics [17], so that considering such a configuration seems interesting and important. This is what we do in the present section.

### 3.1 Green's function in a Fabry-Pérot cavity

Interaction of atoms via the electromagnetic field is expressed in the Hamitonian (1) with the help of the dyadic Green's function $\hat{\mathcal{G}}(\mathbf{r}, \mathbf{r}')$. Section 2 deals with this Hamiltonian for atoms in the free 3D space and makes use of explicit expressions (2) and (16) for $\hat{\mathcal{G}}(\mathbf{r}, \mathbf{r}') = \hat{\mathcal{G}}(\mathbf{r} - \mathbf{r}')$ and its Fourier transform $\hat{\mathcal{G}}(\mathbf{q})$, respectively. It is easy to convince oneself that the same Hamiltonian (1) still holds if the atoms are placed in a different environment that modifies the Green's function. The analysis of Sec. 2 then has to be repeated with a different $\hat{\mathcal{G}}(\mathbf{r}, \mathbf{r}')$.

We assume that a 2D atomic lattice is placed in the middle of a Fabry-Pérot cavity of spacing $d$ between perfectly reflecting mirrors at $z = \pm d/2$. For TE excitations, the electric field is parallel to the mirrors and should vanish on them (we assume that the mirrors are made of a perfect electric conductor material). For a point source at $\mathbf{r}' = \{\boldsymbol{\rho}', z' = 0\}$ in the atomic plane, these boundary conditions can be satisfied by the image method. Namely, we place fictitious point sources of alternating signs at $\mathbf{r}'_n = \{\boldsymbol{\rho}', nd\} = \mathbf{r}' + nd\mathbf{e}_z$ and write

$$\hat{\mathcal{G}}_{\mathrm{FP}}(\mathbf{r}, \mathbf{r}') = \sum_{n=-\infty}^{\infty} (-1)^n \hat{\mathcal{G}}(\mathbf{r}, \mathbf{r}'_n) \tag{29}$$

where we use a subscript 'FP' to distinguish the Green's function in a Fabry-Pérot (FP) cavity from that in the free space and the term corresponding to $n = 0$ accounts for the real source at $\mathbf{r}'_0 = \mathbf{r}'$. Note that only the leading principal submatrix of order 2 of the $3 \times 3$ matrix defined by Eq. (29)—the one describing coupling between in-plane $x$ and $y$ components of atomic dipole moments—makes physical sense because different boundary conditions have to be applied for the $z$ component. It is the only part of $\hat{\mathcal{G}}_{\mathrm{FP}}$ that will be used in what follows. In general, the cavity breaks the translational invariance and $\mathcal{G}_{\mathrm{FP}}$ is not a function of a single variable $\mathbf{r} - \mathbf{r}'$ anymore. However, the translational invariance still exists in the atomic plane $z = 0$ and $\mathcal{G}_{\mathrm{FP}}$ becomes a function of $\mathbf{r} - \mathbf{r}'$ only when both $\mathbf{r}$ and $\mathbf{r}'$ are in the plane $z = 0$.

The Fourier transform of Eq. (29) is

$$\begin{aligned} \hat{\mathcal{G}}_{\mathrm{FP}}(\mathbf{q}, \mathbf{r}') &= \int d^3\mathbf{r}\, \hat{\mathcal{G}}_{\mathrm{FP}}(\mathbf{r}, \mathbf{r}') e^{i\mathbf{q}\cdot\mathbf{r}} = \sum_{n=-\infty}^{\infty} (-1)^n \int d^3\mathbf{r}\, \hat{\mathcal{G}}(\mathbf{r}, \mathbf{r}'_n) e^{i\mathbf{q}\cdot\mathbf{r}} \\ &= \sum_{n=-\infty}^{\infty} (-1)^n \left[ \int d^3\Delta\mathbf{r}_n\, \hat{\mathcal{G}}(\Delta\mathbf{r}_n) e^{i\mathbf{q}\cdot\Delta\mathbf{r}_n} \right] e^{i\mathbf{q}\cdot\mathbf{r}'_n} = \sum_{n=-\infty}^{\infty} (-1)^n \hat{\mathcal{G}}(\mathbf{q}) e^{i\mathbf{q}\cdot\mathbf{r}'_n} \end{aligned} \tag{30}$$

where $\Delta\mathbf{r}_n = \mathbf{r} - \mathbf{r}'_n$.

Equation (18) becomes

$$
\begin{aligned}
\hat{g}_{\text{FP}}^{(\text{reg})}(\mathbf{q}_\perp) &= \int \frac{dq_z}{2\pi} \hat{G}_{\text{FP}}(\mathbf{q}, \mathbf{r}' = \mathbf{0}) e^{-h^2 q^2/2} = \sum_{n=-\infty}^{\infty} (-1)^n \int \frac{dq_z}{2\pi} \hat{G}(\mathbf{q}) e^{iq_z nd} e^{-h^2 q^2/2} \\
&= \hat{g}^{(\text{reg})}(\mathbf{q}_\perp) + 2\sum_{n=1}^{\infty} (-1)^n \int \frac{dq_z}{2\pi} \hat{G}(\mathbf{q}) \cos(q_z nd) \\
&= \hat{g}^{(\text{reg})}(\mathbf{q}_\perp) - \frac{6\pi}{k_0} \sum_{n=1}^{\infty} (-1)^n \frac{e^{-nd\sqrt{q_\perp^2 - k_0^2}}}{\sqrt{q_\perp^2 - k_0^2}} \left[ \mathbb{1}_2 - \frac{\hat{d}_{2D}(\mathbf{q}_\perp \otimes \mathbf{q}_\perp)\hat{d}_{2D}^\dagger}{k_0^2} \right] \\
&= \hat{g}^{(\text{reg})}(\mathbf{q}_\perp) + \frac{6\pi}{k_0} \frac{1}{1 + e^{k_0 d\sqrt{(q_\perp/k_0)^2 - 1}}} \times \frac{\mathbb{1}_2 - \left[\hat{d}_{2D}(\mathbf{q}_\perp \otimes \mathbf{q}_\perp)\hat{d}_{2D}^\dagger\right]/k_0^2}{\sqrt{q_\perp^2 - k_0^2}} \quad (31)
\end{aligned}
$$

where we separate the term corresponding to $n = 0$ and yielding the free-space result $\hat{g}^{(\text{reg})}(\mathbf{q}_\perp)$ given by Eq. (18) and use the fact that $\sum_{n=1}^{\infty}(-1)^n e^{-nx} = -1/(1 + e^x)$.

In its turn, Eq. (19) becomes

$$
\begin{aligned}
\hat{G}_{\text{FP}}^{(\text{reg})}(\mathbf{r} = \mathbf{0}) &= \int \frac{d^3\mathbf{q}}{(2\pi)^3} \hat{G}_{\text{FP}}(\mathbf{q}, \mathbf{r}' = \mathbf{0}) e^{-h^2 q^2/2} = \sum_{n=-\infty}^{\infty} (-1)^n \int \frac{d^3\mathbf{q}}{(2\pi)^3} \hat{G}(\mathbf{q}) e^{iq_z nd} e^{-h^2 q^2/2} \\
&= \hat{G}^{(\text{reg})}(\mathbf{r} = \mathbf{0}) + 2\sum_{n=1}^{\infty} (-1)^n \int \frac{d^3\mathbf{q}}{(2\pi)^3} \hat{G}(\mathbf{q}) \cos(q_z nd) \\
&= \hat{G}^{(\text{reg})}(\mathbf{r} = \mathbf{0}) - \frac{6\pi}{k_0} \sum_{n=1}^{\infty} (-1)^n \int \frac{d^2\mathbf{q}_\perp}{(2\pi)^2} \frac{e^{-nd\sqrt{q_\perp^2 - k_0^2}}}{\sqrt{q_\perp^2 - k_0^2}} \left[ \mathbb{1}_2 - \frac{\hat{d}_{2D}(\mathbf{q}_\perp \otimes \mathbf{q}_\perp)\hat{d}_{2D}^\dagger}{k_0^2} \right] \\
&= \hat{G}^{(\text{reg})}(\mathbf{r} = \mathbf{0}) - \mathbb{1}_2 \times 3 \sum_{n=1}^{\infty} (-1)^n \frac{nk_0 d(nk_0 d - i) - 1}{(nk_0 d)^3} e^{-ink_0 d} \\
&= \hat{G}^{(\text{reg})}(\mathbf{r} = \mathbf{0}) + \mathbb{1}_2 \times 3 \left[ \frac{1}{k_0 d} \ln\left(1 + e^{-ik_0 d}\right) \right. \\
&+ \left. \frac{i}{(k_0 d)^2} \text{Li}_2\left(-e^{-ik_0 d}\right) + \frac{1}{(k_0 d)^3} \text{Li}_3\left(-e^{-ik_0 d}\right) \right] \quad (32)
\end{aligned}
$$

where $\text{Li}_n(z) = \sum_{k=1}^{\infty} z^k/k^n$ is the polylogarithm function and the free-space result $\hat{G}^{(\text{reg})}(\mathbf{r} = \mathbf{0})$ obtained from $n = 0$ term in the sum, is separated from the rest of the equation. Note that we do not need the Gaussian cut-off in the integrals for $n \neq 0$ in Eqs. (31) and (32) where we therefore put $h = 0$.

## 3.2 Atom in a Fabry-Pérot cavity

Before studying collective excitations in a lattice of many atoms inside a Fabry-Pérot cavity, it is instructive to recall some of the results concerning the impact of a cavity on the properties of a single atom. The impact of environment, including mirrors, on the transition frequency and the spontaneous decay rate of atoms has been extensively studied in the framework of cavity quantum electrodynamics [27] even though the problem is of essentially classical nature [28]. Experimental results are available since many years [29, 30]. In fact, the real and imaginary parts of the Green's function at the origin $\hat{G}_{\text{FP}}^{(\text{reg})}(\mathbf{r} = \mathbf{0})$ given by Eq. (32) yield the frequency shift of the atomic resonance $\Delta\omega$ and the decay rate $\Gamma$ of an atom in the presence of the cavity:

$$
\frac{\Delta\omega}{\Gamma_0} = -\frac{1}{2}\text{Re}\left[ \hat{G}_{\text{FP}}^{(\text{reg})}(\mathbf{r} = \mathbf{0}) - \hat{G}^{(\text{reg})}(\mathbf{r} = \mathbf{0}) \right] \quad (33)
$$

$$
\frac{\Gamma}{\Gamma_0} = \text{Im}\left[ \hat{G}_{\text{FP}}^{(\text{reg})}(\mathbf{r} = \mathbf{0}) \right] \quad (34)
$$

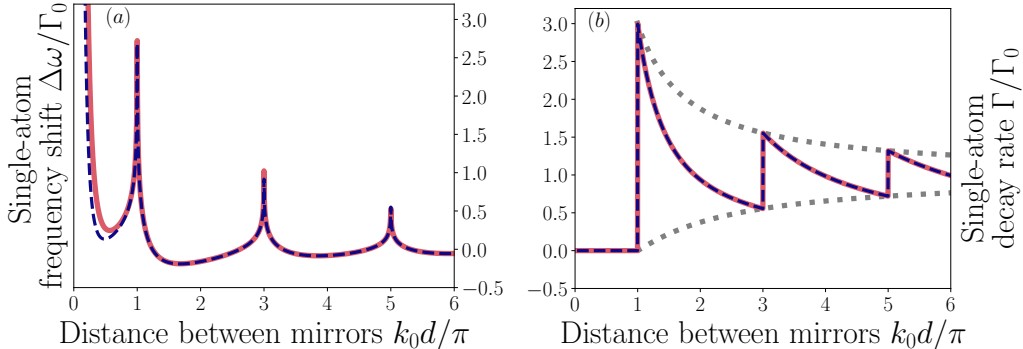

Figure 7: Shift of the resonance frequency (a) and decay rate of the excited state (b) of a single atom in the middle of a Fabry-Pérot cavity for the dipole moment of the atomic transition parallel to the mirrors (red solid lines). Dashed lines show results of Ref. [31]. Dotted lines in (b) show Eq. (35).

We show the dependence of $\Delta\omega$ and $\Gamma$ on the distance $d$ between cavity mirrors in Fig. 7. Both $\Delta\omega$ and $\Gamma$ exhibit sharp variations at $k_0 d = (2n+1)\pi$ with integer $n = 0, 1, \ldots$ due to the sudden emergence of new TE modes in the Fabry-Pérot cavity at these values of $k_0 d$. New modes also appear at $k_0 d = 2n\pi$ but they provoke no effect because they have nodes in the plane $z = 0$ where the atom is located. Thus, the atom does not couple to these modes. $\Delta\omega$ remains small except for the limit $d \to 0$ and for logarithmic divergences at $k_0 d = (2n+1)\pi$. The modification of $\Gamma$ by the cavity is more important. First, $\Gamma = 0$ for $k_0 d < \pi$. No TE modes exist in the cavity for such small $d$ and thus the atom cannot emit a photon and remains excited for an infinitely long time $1/\Gamma \to \infty$. For $k_0 d = (2n+1)\pi$, $\Gamma$ exhibits sharp jumps between values shown by dotted lines in Fig. 7(b):

$$\left.\frac{\Gamma}{\Gamma_0}\right|_{k_0 d = (2n+1)\pi \pm 0^+} = 1 + \frac{1}{2}\left[\frac{1}{(2n+1)^2} \pm \frac{3}{2n+1}\right] \tag{35}$$

The values of $\Gamma/\Gamma_0$ for $k_0 d = 2n\pi$ are close to 1:

$$\left.\frac{\Gamma}{\Gamma_0}\right|_{k_0 d = 2n\pi} = 1 - \frac{1}{(4n)^2} \tag{36}$$

As could be expected, the impact of the cavity decreases and $\Delta\omega \to 0$, $\Gamma \to \Gamma_0$ as $d$ increases.

Figure 7 also compares our results with a previous solution to the same problem by Milonni and Knight [31] shown by dashed lines.[1] The results for $\Gamma$ coincide exactly whereas those for $\Delta\omega$ exhibit a slight discrepancy for $k_0 d < \pi$. Close examination allows us to conclude that the discrepancy arises from the neglect of the longitudinal electromagnetic fields in Ref. [31]. Indeed, we are able to reproduce the dashed line in Fig. 7 by repeating our calculations without the longitudinal part $(\mathbf{q} \otimes \mathbf{q})/(q^2 k_0^2)$ of the dyadic Green's function (16).

### 3.3 Band diagram of a honeycomb lattice in a Fabry-Pérot cavity

To obtain a band diagram of the honeycomb atomic lattice placed in the middle of a Fabry-Pérot cavity, we diagonalize the $4 \times 4$ Hamiltonian (8) with entries given by Eqs. (9)–(12) where

---

[1]For the atomic transition dipole moment parallel to the cavity mirrors, Ref. [31] provides an explicit expression only for the decay rate but not for the frequency shift. We derived the latter using the same formalism and by analogy with the case of the dipole moment perpendicular to the mirrors that is treated in more detail in that work.

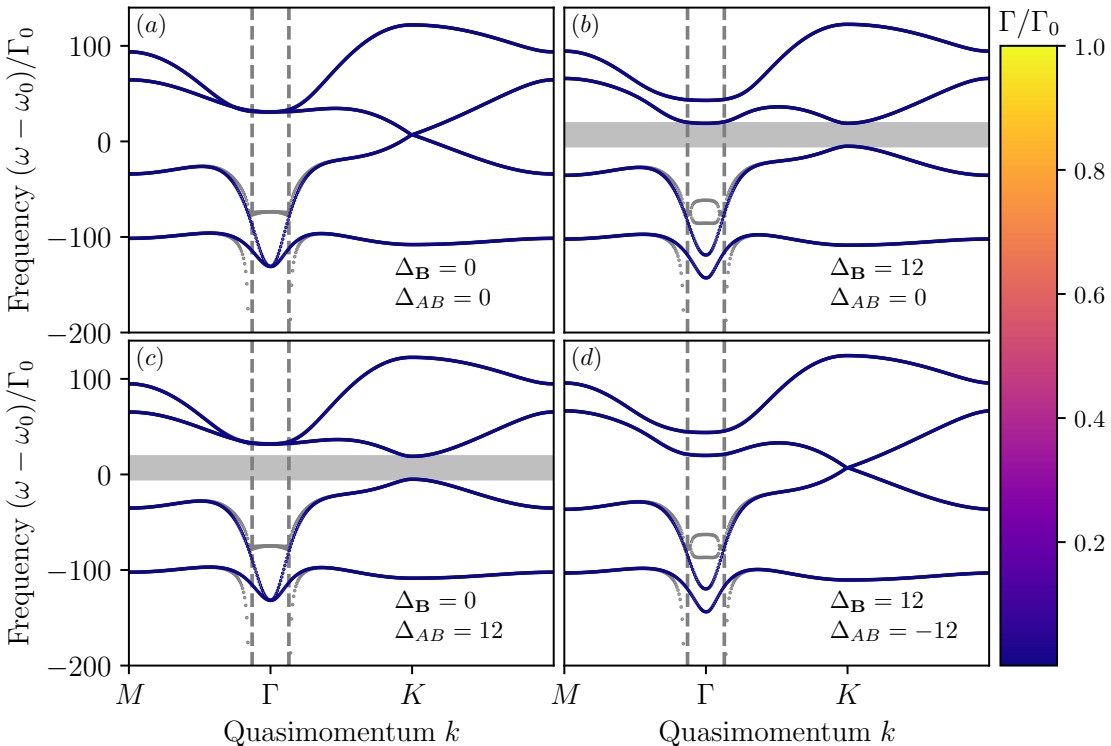

Figure 8: Same as Fig. 2 but for a honeycomb atomic lattice placed in the middle of a Fabry-Pérot cavity of width $k_0 d = 2$. The band diagrams of Fig. 2 are also shown for comparison (gray points).

the sums over $\mathbf{r}_n$ are calculated using Eqs. (13) and (17) with $\hat{g}(\mathbf{q}_\perp)$ and $\hat{G}(\mathbf{r} = 0)$ replaced by $\hat{g}_{\mathrm{FP}}^{(\mathrm{reg})}(\mathbf{q}_\perp)$ and $\hat{G}_{\mathrm{FP}}^{(\mathrm{reg})}(\mathbf{r} = 0)$ given by Eqs. (31) and (32), respectively. In addition, we account for the modification of single-atom transition frequency and decay rate discussed in Sec. 3.2 by replacing $i$ on the diagonals of the second terms on the right-hand sides of Eqs. (9) and (12) by $\hat{G}_{\mathrm{FP}}^{(\mathrm{reg})}(\mathbf{r} = 0) - \mathrm{Re}\hat{G}^{(\mathrm{reg})}(\mathbf{r} = 0)$. The resulting band diagrams are shown in Figs. 8 and 9 for $k_0 d = 2$ and 11, respectively, and the same values of $k_0 a$, $\Delta_{\mathbf{B}}$ and $\Delta_{AB}$ as in Fig. 2.

The first observation following from Figs. 8 and 9 is that the imaginary part of the eigenvalues $\Lambda(\mathbf{k})$ of $\hat{\mathcal{H}}(\mathbf{k})$ vanishes in the presence of the Fabry-Pérot cavity. In fact, the numerical diagonalization of $\hat{\mathcal{H}}(\mathbf{k})$ still yields $\Lambda(\mathbf{k})$ with small but non-zero imaginary parts for some values of $\mathbf{k}$, but we can show that this is due to the insufficiently small cut-off lengths $h$ used to compute the regularized Green's functions (18) and (19). Decreasing $h$ reduces $\mathrm{Im}\Lambda(\mathbf{k}) = \Gamma(\mathbf{k})/\Gamma_0$ and we find a relation $\max_{\mathbf{k}}[\Gamma(\mathbf{k})/\Gamma_0] \propto (k_0 h)^2$ as illustrated in Fig. 10(a). Thus, $\Gamma(\mathbf{k}) = 0$ in the physically relevant limit $h \to 0$.

The second observation following from Figs. 8 and 9 is that the band diagrams of the atomic lattice inside a cavity are very similar to the one for the same lattice in the free space, except for the vicinity of $\Gamma$ point $|\mathbf{k}| < k_0$. It might be expected that the cavity has the strongest impact inside this light cone because this is where the emission of electromagnetic waves into the free space is suppressed by the cavity. However, it is less obvious that the cavity has virtually no effect outside the light cone.

Figure 8 is obtained for $k_0 d < \pi$, when the cavity does not support propagating TE modes. Thus, the atoms are coupled by near fields only and the coupling is necessarily short-ranged. The fact that the band diagram is still similar to the one obtained for the atomic lattice in the free space (at least, for $|\mathbf{k}| > k_0$) even under such extreme conditions, shows that the near-

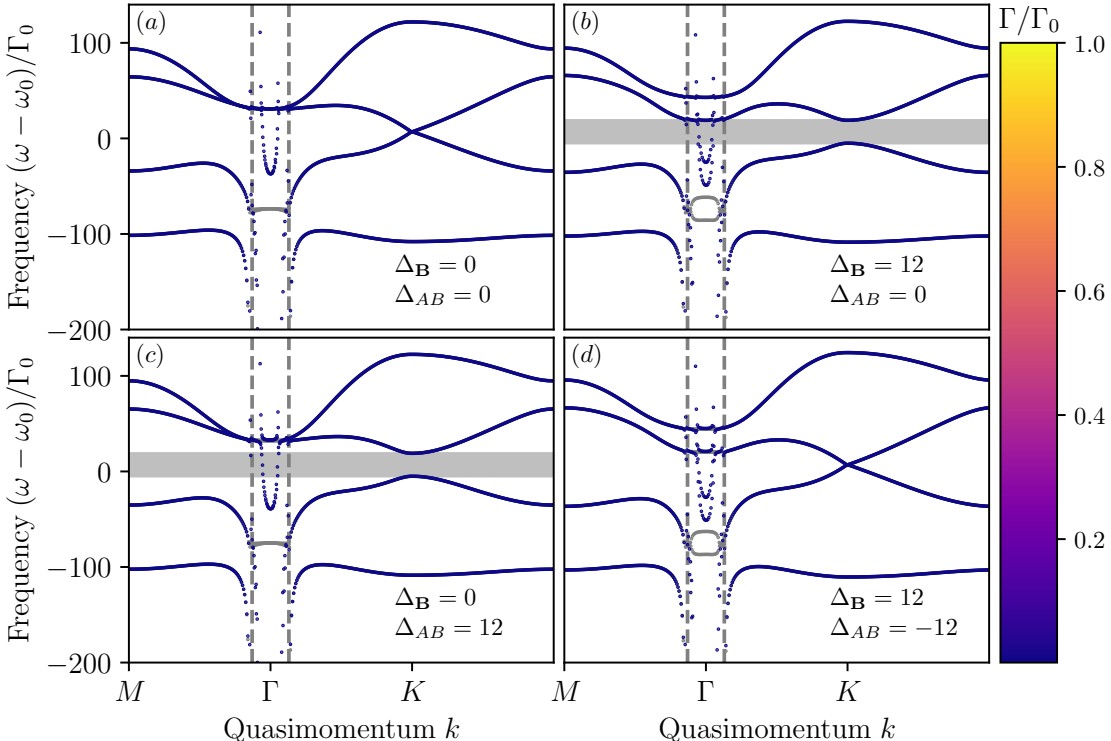

Figure 9: Same as Fig. 8 but for $k_0 d = 11$. The band diagrams of Fig. 2 are also shown for comparison (gray points in the background).

field coupling plays the central role even in the free space, despite the presence of TE modes able to couple distant atoms. Note that Fig. 8 does not exhibit divergences of $\omega$ at $|\mathbf{k}| = k_0$, in contrast to Fig. 2 in the free space. This is also due to the absence of long-range coupling between atoms.

For large enough distances $d$ between mirrors of the Fabry-Pérot cavity, the band diagram becomes identical to that of the atomic lattice in the free space for all $|\mathbf{k}| > k_0$ but very different from the latter inside the light cone $|\mathbf{k}| < k_0$ (see Fig. 9 for $k_0 d = 11$). The main difference is, of course, that $\Gamma = 0$ for all $\mathbf{k}$, even for large $d$. Instead of carrying energy out of the atomic system and giving rise to large $\Gamma$, modes with $|\mathbf{k}| < k_0$ are now stuck in between the cavity mirrors and feature very steep dispersion curves $\omega(\mathbf{k})$ diverging at $|\mathbf{k}| = k_0$ and evolving rapidly with $k_0 d$. To understand the origin of these new modes, we note that the spectrum of excitations of a 2D atomic lattice in the middle of a Fabry-Pérot cavity is identical to that of a periodic stack of identical lattices along $z$ axis, with a spacing $d$ and with alternating signs of the wave function $\mathbf{\Psi}$ in consequent atomic planes. The modes inside the light cone $|\mathbf{k}| < k_0$ in Fig. 9 are those involving different atomic planes whereas the modes outside the cone are confined in individual planes. The modes inside the light cone close spectral gaps in Fig. 9(b) and (c) and may make impossible observation of topological properties in an experiment, unless the experiment is designed to probe the vicinity of $K$ point and to be insensitive to the processes involving excitations in the vicinity of $\Gamma$ point.

## 3.4 Topological properties of the band structure

To explore the topological properties of the band structures obtained for the atomic lattice placed in a Fabry-Pérot cavity, we repeat calculations described in Sec. 2.4. We find that the conclusions reached in Sec. 2.4 still hold for the lattice in the cavity as long as $k_0 d < \pi$.

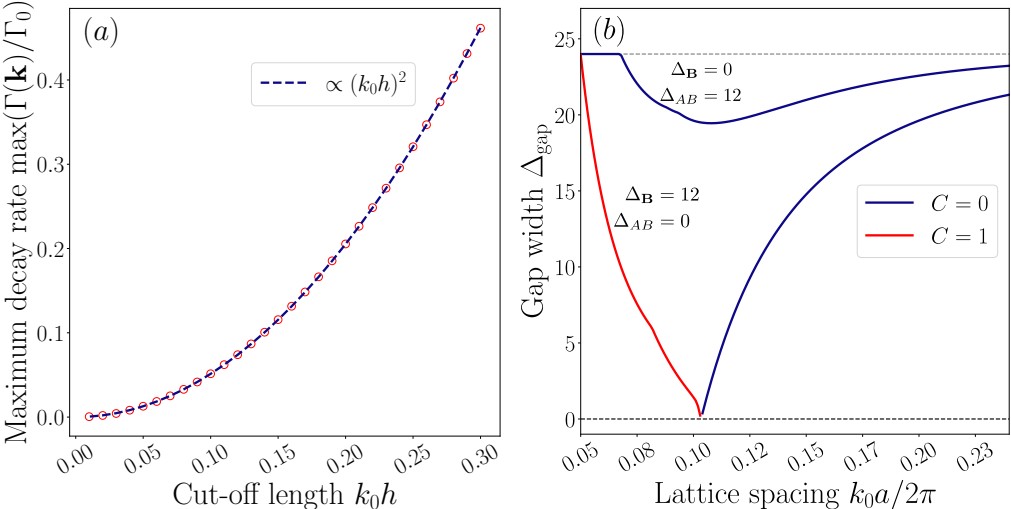

Figure 10: (a) Maximum decay rate for the band diagram of Fig. 8 as a function of cut-off length $h$ (red circles). Blue dashed line shows a parabolic fit to the numerical data. (b) Width of the gap between the second and third bands of the spectrum of a honeycomb atomic lattice in the middle of a Fabry-Pérot cavity ($k_0 d = 2$), as a function of lattice spacing $a$, for a system that is topologically trivial (upper line) or not (lower line). Color code shows Chern number of the gap: $C = 0$ (blue) or $C = 1$ (red). Horizontal dashed line shows the gap width in the limit $a \to \infty$.

For larger $k_0 d$, the calculation of the Chern number is complicated by the modes inside the light cone that close the gap. We thus focus on $k_0 d < \pi$ and further demonstrate the difference between trivial (for $|\Delta_{AB}| > |\Delta_{\mathbf{B}}|$) and topological (for $[\Delta_{AB}| < |\Delta_{\mathbf{B}}|$) gaps by studying the evolution of the gap width and of its Chern number with the interatomic spacing $a$. Four eigenvalues of $\hat{\mathcal{H}}$ in the limit of $a \to \infty$ readily follow from Eqs. (8) and (33): $\Lambda = \mathrm{Re}\left[\hat{G}_{\mathrm{FP}}^{(\mathrm{reg})}(\mathbf{r}=0) - \hat{G}^{(\mathrm{reg})}(\mathbf{r}=0)\right] \pm 2\Delta_{AB} \pm 2\Delta_{\mathbf{B}}$. When $\Delta_{\mathbf{B}} = 0$ or $\Delta_{AB} = 0$, there are two pairs of doubly degenerate eigenvalues $\Lambda = \mathrm{Re}\left[\hat{G}_{\mathrm{FP}}^{(\mathrm{reg})}(\mathbf{r}=0) - \hat{G}^{(\mathrm{reg})}(\mathbf{r}=0)\right] \pm 2\Delta_{AB}$ or $\Lambda = \mathrm{Re}\left[\hat{G}_{\mathrm{FP}}^{(\mathrm{reg})}(\mathbf{r}=0) - \hat{G}^{(\mathrm{reg})}(\mathbf{r}=0)\right] \pm 2\Delta_{\mathbf{B}}$, respectively. The band gap $\Delta_{\mathrm{gap}} = 2|\Delta_{AB}|$ or $2|\Delta_{\mathbf{B}}|$ between the corresponding frequencies is, of course, topologically trivial. An interesting question is how this trivial limit can be reached starting from the dense atomic lattice with small $a$. A continuous transition from small $a$ to large $a$ is illustrated in Fig. 10(b). When the gap is trivial ($C = 0$) for small $a$, increasing $a$ leads to a slight reduction of the gap width that finally converges to its value in the lattice with an infinitely large spacing $a$ shown by a dashed line in the figure. In contrast, when the gap is topological ($C = 1$ for small $a$), the gap width decreases rapidly with $a$ until a complete gap closing [for $k_0 a/2\pi \simeq 0.1$ in Fig. 10(b)]. Further increase of $a$ reopens a topologically trivial gap ($C = 0$) with a width slowly converging to its infinite-$a$ limit as $a$ increases.

Figure 10(b) provides a nice visual illustration of the fact that different states of a physical system (here, the many-body state for small $a$ and the two-atom state for infinite $a$) that are characterized by the same value of a topological invariant (here, Chern number $C = 0$), can be connected by a path in the parameter space (here, the path goes along the axis $a$ at constant $\Delta_{AB}$ and $\Delta_{\mathbf{B}}$) that keeps a spectral gap always open. In contrast, a path connecting two states characterized by different values of $C$ ($C = 1$ for small $a$ and $C = 0$ for large $a$) necessarily implies closing of the gap.

# 4 Conclusion

We provide a complete characterization of the band structure of a 2D honeycomb lattice of atoms interacting via the electromagnetic field polarized in the plane of the lattice (TE modes). Atoms are assumed to have a nondegenerate ground state (total angular momentum $J_g = 0$) and a triply degenerate excited state ($J_e = 1$), which leads to a band structure composed of four bands. The lattice embedded in the free 3D space and the lattice placed in the middle of a Fabry-Pérot cavity of spacing $d < \pi/k_0$ have very similar properties, except for the large decay rates $\Gamma$ of modes of the lattice in the free space for 2D wave vectors $\mathbf{k}$ inside the light cone $|\mathbf{k}| < k_0$. The cavity blocks the emission of electromagnetic energy in the infinite free space, leading to $\Gamma = 0$ for all modes. If the lattice spacing $a$ is small enough ($k_0 a \lesssim 0.1$), a gap can be opened between the second and third spectral bands by breaking either the time-reversal (by an external magnetic field leading to a Zeeman shift $\Delta_\mathbf{B}$ in units of the decay rate of the excited state $\Gamma_0$) or the inversion (by assigning different resonant frequencies $\omega_0 \mp \Delta_{AB}\Gamma_0$ to atoms $A$ and $B$ forming the two-atom unit cell of the honeycomb lattice) symmetry. We characterize the topological property of the spectral gap by its Chern number $C$. The gap is topological ($C \neq 0$) when the breakdown of the time-reversal symmetry is stronger than the breakdown of the inversion symmetry: $|\Delta_\mathbf{B}| > |\Delta_{AB}|$. The gap is trivial ($C = 0$) in the opposite case. The topological character of the gap for $|\Delta_\mathbf{B}| > |\Delta_{AB}|$ leads to a characteristic band inversion: the population disbalance between $A$ and $B$ sublattices for states near band edges is opposite to that in the case of a trivial band gap. In addition, reaching the trivial limit of noninteracting atoms by increasing the lattice spacing $a$ requires closing of the topological gap and then reopening of a trivial one, in contrast to the case of the topologically trivial gap that remains open all the way from small to large $a$. The topological gap $\Delta_\mathrm{gap}$ is the largest for a range of $|\Delta_\mathbf{B}|$ between two values determined by the dimensionless lattice spacing $k_0 a$ and the inversion symmetry breaking strength $|\Delta_{AB}|$. Near-field dipole-dipole interactions between atoms play a crucial role in the opening of the topological spectral gap, leading to its maximum width $\max(\Delta_\mathrm{gap}) \propto (k_0 a)^{-3}$ for $\Delta_{AB} = 0$.

The band diagram of the atomic lattice placed in the middle of a Fabry-Pérot cavity of spacing $d > \pi/k_0$ differs form the diagram for $d < \pi/k_0$ only slightly. The difference is concentrated inside the light cone $|\mathbf{k}| < k_0$ where new modes arise due to reflections of light from cavity mirrors. These new modes still have $\Gamma = 0$, so that the spectrum remains real-valued. They close the band gap that would be present for $d < \pi/k_0$ and complicate the study of the topological properties of the band diagram.

We believe that the recent progress in manipulating cold atoms and, in particular, the remarkable achievements on the way towards arranging them in regular lattices with lattice spacings well below the optical wavelength [32–34], give our theoretical results a direct experimental relevance. In addition, our theoretical model can also be used for an approximate description of electromagnetic wave propagation in 2D arrays of small dielectric scatterers (or pillars or rods, to use the standard terminology), both in microwave and optical spectral ranges, at frequencies near an isolated single-scatterer resonance of electric-dipolar nature. Topological photonics of arrays of dielectric resonators is an active research field (see Refs. [17, 22–25] for recent reviews). Even though in most of the existing works one relies on isotropic $s$-modes of individual pillars, corresponding to magnetic-dipolar resonances, $p$-modes and the corresponding electric-dipolar resonances have also been studied in the literature [17, 35]. For such resonances, considering a pillar as point-like would allow for using the theoretical framework developed in this work to obtain a description of a lattice composed of many identical pillars that become "atoms" of our model. Though only approximate, such a description should be much less computer-resource consuming than the direct numerical solution of Maxwell equations and may be helpful in understanding the basic physical processes

at play.

**Funding information**    This work was funded by the Agence Nationale de la Recherche (Grant No. ANR-20-CE30-0003 LOLITOP).

**Source code**    The code used to generate numerical results and figures will be made available upon publication.

# A    Derivation of the formula for the width of the spectral gap

Analysis of band diagrams $\omega_\alpha(\mathbf{k})$ of the Hamiltonian $\hat{\mathcal{H}}(\mathbf{k})$ given by Eqs. (8)–(12) shows that the width of the gap between the second and third bands is determined by the values of $\omega_2$ and $\omega_3$ at very specific points of BZ: $\Gamma$, $K$ and $K$' points. At these points, $\hat{\mathcal{H}}(\mathbf{k})$ takes quite simple forms facilitating its diagonalization. In particular, at $K$ point we have

$$\hat{\mathcal{H}}(\mathbf{k}_K) = c_0 \mathbb{1}_4 + 2 \begin{pmatrix} \Delta_{AB} + \Delta_{\mathbf{B}} & 0 & 0 & \Omega \\ 0 & \Delta_{AB} - \Delta_{\mathbf{B}} & 0 & 0 \\ 0 & 0 & -\Delta_{AB} + \Delta_{\mathbf{B}} & 0 \\ \Omega & 0 & 0 & -\Delta_{AB} - \Delta_{\mathbf{B}} \end{pmatrix} \tag{37}$$

where

$$\begin{aligned} c_0 &= i + \sum_{\mathbf{r}_n \neq 0} G_{++}(\mathbf{r}_n) e^{i\mathbf{k}_K \cdot \mathbf{r}_n} = i + \frac{1}{\mathcal{A}} \sum_{\mathbf{g}_m} g_{++}(\mathbf{g}_m - \mathbf{k}_K) - G_{++}(\mathbf{r}=0) \\ &= \frac{1}{\mathcal{A}} \sum_{\mathbf{g}_m} g_{++}(\mathbf{g}_m - \mathbf{k}_K) - \mathrm{Re}\, G_{++}(\mathbf{r}=0) \end{aligned} \tag{38}$$

$$\Omega = \sum_{\mathbf{r}_n} G_{+-}(\mathbf{r}_n + \mathbf{a}_1) e^{i\mathbf{k}_K \cdot \mathbf{r}_n} = \frac{1}{\mathcal{A}} \sum_{\mathbf{g}_m} g_{+-}(\mathbf{g}_m - \mathbf{k}_K) e^{i\mathbf{a}_1 \cdot (\mathbf{g}_m - \mathbf{k}_K)} \tag{39}$$

are functions of $k_0 a$ and $\mathbf{k}_K = \{K, 0\}$ is the value of $\mathbf{k} = \{k_x, k_y\}$ at the $K$ point of BZ with $K = 4\pi/3\sqrt{3}a$. For $k_0 a \ll 1$, we find $|\Omega| \gg |c_0|$. The four eigenvalues of $\hat{\mathcal{H}}(\mathbf{k}_K)$ are real:

$$\Lambda^K = c_0 \pm 2\sqrt{(\Delta_{\mathbf{B}} - \Delta_{AB})^2 + \Omega^2/4} \simeq c_0 \pm \Omega \tag{40}$$

and

$$\Lambda^K = c_0 \pm 2(\Delta_{AB} + \Delta_{\mathbf{B}}) \tag{41}$$

The result at $K'$ point $\mathbf{k}'_K = -\mathbf{k}_K$ is obtained by changing the sign in front of $\Delta_{\mathbf{B}}$ in the above expressions.

     At the $\Gamma$ point $\mathbf{k}_\Gamma = 0$ we find

$$\hat{\mathcal{H}}(0) = (c_1 + ic_3)\mathbb{1}_4 + 2 \begin{pmatrix} \Delta_{AB} + \Delta_{\mathbf{B}} & 0 & (c_2 + ic_3)/2 & 0 \\ 0 & \Delta_{AB} - \Delta_{\mathbf{B}} & 0 & (c_2 + ic_3)/2 \\ (c_2 + ic_3)/2 & 0 & -\Delta_{AB} + \Delta_{\mathbf{B}} & 0 \\ 0 & (c_2 + ic_3)/2 & 0 & -\Delta_{AB} - \Delta_{\mathbf{B}} \end{pmatrix} \tag{42}$$

where $c_1$, $c_2$ and $c_3$ are real-valued functions of $k_0 a$ and are defined by the following relations:

$$c_1 + ic_3 = i + \sum_{\mathbf{r}_n \neq 0} G_{++}(\mathbf{r}_n) = i + \frac{1}{\mathcal{A}} \sum_{\mathbf{g}_m} g_{++}(\mathbf{g}_m) - G_{++}(\mathbf{r} = 0)$$

$$= \frac{1}{\mathcal{A}} \sum_{\mathbf{g}_m} g_{++}(\mathbf{g}_m) - \mathrm{Re}\, G_{++}(\mathbf{r} = 0) \tag{43}$$

$$c_2 + ic_3 = \sum_{\mathbf{r}_n} G_{++}(\mathbf{r}_n + \mathbf{a}_1) = \frac{1}{\mathcal{A}} \sum_{\mathbf{g}_m} g_{++}(\mathbf{g}_m) e^{i\mathbf{a}_1 \cdot \mathbf{g}_m} \tag{44}$$

The four complex eigenvalues of $\hat{\mathcal{H}}(0)$ are given by

$$\Lambda_\Gamma = c_1 + ic_3 \pm 2\Delta_{\mathbf{B}} \pm 2\sqrt{\Delta_{AB}^2 + (c_2 + ic_3)^2/4} \tag{45}$$

with the four possible combinations of $\pm$ signs.

Further analysis reduces to following the eigenvalues $\Lambda_K$, $\Lambda_{K'}$ and the real part of $\Lambda_\Gamma$ as $\Delta_{\mathbf{B}}$ varies at fixed $k_0 a$ and $\Delta_{AB}$. At each value of $\Delta_{\mathbf{B}}$, we sort $\Lambda_K$ and $\mathrm{Re}\Lambda_\Gamma$ in descending order and find the width of the gap between the second and third bands as

$$\Delta_{\mathrm{gap}} = \frac{1}{2} \min \left\{ \Lambda_K^{(2)} - \Lambda_K^{(3)}, \Lambda_K^{(2)} - \mathrm{Re}\Lambda_\Gamma^{(3)}, \mathrm{Re}\Lambda_\Gamma^{(2)} - \Lambda_K^{(3)}, \mathrm{Re}\Lambda_\Gamma^{(3)} - \mathrm{Re}\Lambda_\Gamma^{(2)}, \right.$$
$$\left. \Lambda_{K'}^{(2)} - \Lambda_{K'}^{(3)}, \Lambda_{K'}^{(2)} - \mathrm{Re}\Lambda_\Gamma^{(3)}, \mathrm{Re}\Lambda_\Gamma^{(2)} - \Lambda_{K'}^{(3)} \right\} \tag{46}$$

As a result of such an analysis, we find that the width of the spectral gap depends only on the absolute values of $\Delta_{\mathbf{B}}$ and $\Delta_{AB}$ and identify three threshold values of $|\Delta_{\mathbf{B}}|$ at which the functional dependence of $\Delta_{\mathrm{gap}}$ on parameters changes because a different term starts to control the minimum in Eq. (46):

$$\Delta_{\mathbf{B}}^{(1)} = \frac{1}{4} |c_0 - c_1 + S + 2|\Delta_{AB}|| \tag{47}$$

$$\Delta_{\mathbf{B}}^{(2)} = \frac{1}{4} |c_0 - c_1 - S - 2|\Delta_{AB}|| \tag{48}$$

$$\Delta_{\mathbf{B}}^{(3)} = \frac{S}{2} \tag{49}$$

where

$$S = 2\mathrm{Re}\sqrt{\Delta_{AB}^2 + \frac{1}{4}(c_2 + ic_3)^2} \tag{50}$$

The result for the width of the gap is

$$\Delta_{\mathrm{gap}} = \begin{cases} 2||\Delta_{\mathbf{B}}| - |\Delta_{AB}||, & |\Delta_{\mathbf{B}}| < \Delta_{\mathbf{B}}^{(1)} \\ \left|\frac{1}{2}(c_0 - c_1 + S) - |\Delta_{AB}|\right|, & \Delta_{\mathbf{B}}^{(1)} < |\Delta_{\mathbf{B}}| < \Delta_{\mathbf{B}}^{(2)} \\ S - 2|\Delta_{\mathbf{B}}|, & \Delta_{\mathbf{B}}^{(2)} < |\Delta_{\mathbf{B}}| < \Delta_{\mathbf{B}}^{(3)} \\ 0, & |\Delta_{\mathbf{B}}| > \Delta_{\mathbf{B}}^{(3)} \end{cases} \tag{51}$$

Equations (51) can also be rewritten in terms of $\Delta_{\mathbf{B}}^{(n)}$ only, without using the parameters $c_n$. This is done in Eq. (20) of the main text.

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
