# Peer review of "Topological photonic band gaps in honeycomb atomic arrays"

_SciPost Physics_

## Round 1 · Referee Report · Anonymous (Referee 1) · 2024-2-1

Report

In this work Wulles and Skipetrov discuss the properties of honeycomb atomic lattices and their associated photonic band gaps. They discuss the magnitude and topological properties of these gaps when the arrays are placed in free space or inside a Fabry-Perot cavity.

I believe the paper is sound and it should be published. Perhaps the results are not too surprising, given what we know from topological phases, and so SciPost Physics Core might be a more appropriate journal.

This suggestion, based on the acceptance criteria, does not diminish the usefulness of their results, in my view. Their quantitative results and remarks can guide intuition in experimental realizations of photonic lattices. As an outsider of the photonics field I appreciated the pedagogical tone. I have a few comments for the authors to address.

1. When computing the Chern number in section 2.4 I would like the authors to discuss how the singularities around Gamma affect the Chern number calculation. Do they pose any problems in convergence?

2. In the same section, I am a little surprised that they don’t make connection to the extensive literature of non-Hermitian topology. In particular the system in free space has a clear non-hermitian component due to the decay rate. In non-Hermitian systems one can also define Chern numbers. More generally, when the spectrum is complex, as it is the case up to section 3, one can define invariants in the space of eigenvalues (Re[E] and Im[E]). Here however, the authors choose to define the Chern number as if the decay rate is not zero. I think the paper can improve substantially if the connection with non-hermitian topology is made.

3. Lastly, the authors say at the end that when d< pi/k_0 it is hard to define topology. They also observe that the system becomes gapless. Hence, their remark that this complicates the calculation of insulators seems to be not a well posed question, since to define invariants for insulators one needs a gap. Perhaps they can clarify further what they mean.

  • validity: high
  • significance: good
  • originality: good
  • clarity: top
  • formatting: excellent
  • grammar: excellent

Author:  Pierre Wulles  on 2024-04-04  [id 4392]

(in reply to Report 1 on 2024-02-01)

Dear referee,

Please find attached the response to the report and the list of changes.

Best regards,
P. Wulles

Attachment:

answer_referee_2.pdf

Author:  Pierre Wulles  on 2024-03-27  [id 4380]

(in reply to Report 1 on 2024-02-01)
Category:
answer to question

Thank you for your feedback. Attached is a document that addresses your comments and accompanies the revised version of the article.

Best regards,

Attachment:

answer_referee_2.pdf

---

## Editorial Decision

resubmitted